# Whole-Genome Analysis of *Mycobacterium neoaurum* DSM 1381 and the Validation of Two Key Enzymes Affecting C22 Steroid Intermediates in Sterol Metabolism

**DOI:** 10.3390/ijms24076148

**Published:** 2023-03-24

**Authors:** Jingxian Zhang, Ruijie Zhang, Shikui Song, Zhengding Su, Jiping Shi, Huijin Cao, Baoguo Zhang

**Affiliations:** 1Lab of Biorefinery, Shanghai Advanced Research Institute, Chinese Academy of Sciences, No. 99 Haike Road, Pudong, Shanghai 201210, China; 2University of Chinese Academy of Sciences, Beijing 100049, China; 3BioTechnology Institute, University of Minnesota, 140 Gortner Lab, 1479 Gortner Avenue Saint Paul, Minneapolis, MN 55108, USA; 4Protein Engineering and Biopharmaceutical Sciences Laboratory, Hubei University of Technology, Wuhan 430068, China

**Keywords:** genome sequencing, *Mycobacterium neoaurum*, *hsd4A*, *kshA1*, homology modeling

## Abstract

*Mycobacterium neoaurum* DSM 1381 originated from *Mycobacterium neoaurum* ATCC 25790 by mutagenesis screening is a strain of degrading phytosterols and accumulating important C22 steroid intermediates, including 22-hydroxy-23, 24-bisnorchola-4-en-3-one (4-HP) and 22-hydroxy-23, 24-bisnorchola-1,4-dien-3-one (HPD). However, the metabolic mechanism of these C22 products in *M. neoaurum* DSM 1381 remains unknown. Therefore, the whole-genome sequencing and comparative genomics analysis of *M. neoaurum* DSM 1381 and its parent strain *M. neoaurum* ATCC 25790 were performed to figure out the mechanism. As a result, 28 nonsynonymous single nucleotide variants (SNVs), 17 coding region Indels, and eight non-coding region Indels were found between the genomes of the two strains. When the wild-type 3-ketosteroid-9α-hydroxylase subunit A1 (KshA1) and β-hydroxyacyl-CoA dehydrogenase (Hsd4A) were overexpressed in *M. neoaurum* DSM 1381, the steroids were transformed into the 4-androstene-3, 17- dione (AD) and 1,4-androstadiene-3,17-dione (ADD) instead of C22 intermediates. This result indicated that 173N of KshA1 and 171K of Hsd4A are indispensable to maintaining their activity, respectively. Amino acid sequence alignment analysis show that both N173D in KshA1 and K171E in Hsd4A are conservative sites. The 3D models of these two enzymes were predicted by SWISS-MODEL and AlphaFold2 to understand the inactivation of the two key enzymes. These results indicate that K171E in Hsd4A may destroy the inaction between the NAD+ with the NH3+ and N173D in KshA1 and may disrupt the binding of the catalytic domain to the substrate. A C22 steroid intermediates–accumulating mechanism in *M. neoaurum* DSM 1381 is proposed, in which the K171E in Hsd4A leads to the enzyme’s inactivation, which intercepts the C19 sub-pathways and accelerates the C22 sub-pathways, and the N173D in KshA1 leads to the enzyme’s inactivation, which blocks the degradation of C22 intermediates. In conclusion, this study explained the reasons for the accumulation of C22 intermediates in *M. neoaurum* DSM 1381 by exploring the inactivation mechanism of the two key enzymes.

## 1. Introduction

Steroid drugs have a variety of pharmacological and physiological activities, are widely used in anti-inflammatory and anti-tumor treatments and to regulate sexual ability and birth control [1,2]. Steroid drugs have become the second-largest marketed pharmaceutical after antibiotics, and the market demand for steroid drugs is continuously increasing [2]. C22 intermediates are highly valuable precursors in steroid drugs, for example in synthesizing progestational and adrenocortical hormones, compared with other steroid intermediates [3]. At present, most of the steroid intermediates used in industrial production are chemical modifications from a natural steroid [4]. Compared with traditional chemical synthesis methods, the biotransformation method has the advantages of mild transformation conditions, fewer production steps, higher transformation efficiency, stereo-selectivity and less environmental pollution. Therefore, biotransformation methods are increasingly used in the production of steroid drugs and intermediates and greatly promote the industrial production of steroid intermediates [5]. Most of the strains used in the industrial production of steroid drug intermediates are from mycobacteria. The development of ideal industry strains for diverse steroid intermediates will significantly promote the application of biotransformation in steroid drug production [5,6,7]. Several efficient steroid intermediate producers have been reported in the last decades [2,6,8]. However, the products are limited to a few intermediates, such as C19 intermediates (AD, ADD, and 9 α-OH-AD) and C22 intermediates (4-HP, HPD, and 9 α-OH-HP) [9].

At present, the steroids that can be obtained by one-step fermentation by microbial transformation are mainly C19 intermediates. The accumulation of C22 intermediates is less studied and developed [3,10]. C22 intermediates are mostly derived from the transformation of steroids by natural or mutagenic strains. The accumulation of C22 intermediates is due to the existence of two sub-pathways in the steroid side-chain degradation pathway by microorganisms: one is the complete degradation of the side chain, of which the final product is a C19 steroid intermediate; the other is the incomplete chain degradation, of which the final product is a C22 steroid intermediate. The steroid side chain metabolic pathway branches at the intermediate 22-Hydroxy-3-oxo-cholest-4-ene-24-carboxyl-CoA. Bifunctional enzyme hydroxyacyl-CoA dehydrogenase/17β-hydroxysteroid dehydrogenase Hsd4A catalysis is reported to account for the branches [3]. A common strategy to obtain intermediates including the steroids mentioned above is to retain the steroid nucleus by knocking out 3-ketosteroid-Δ1-dehydrogenase (KstD) or 3-ketosteroid-9α-hydroxylase (KSH) to prevent the ring-opening reaction of the B ring [10].

Some efficient steroid drug intermediate producers have been obtained by traditional mutagenesis [11]. However, it is hard to further reconstruct these strains to achieve higher yield and fewer by-products due to the lack of knowledge of steroid degradation pathways [12]. Genome sequencing is one of the important methods to reveal the steroid degradation mechanism. Until now, several *M. neoaurum* strains have been sequenced, including type strains *M. neoaurum* ATCC 25795 and *M. neoaurum* NRRLB 3805, and industrial steroid-producing strains *Mycobacterium* sp. VKM Ac-1815D and *Mycobacterium* sp. VKM Ac-1816D [13,14]. However, the strains that can accumulate C22 intermediates were not sequenced and studied. Among the various mutants, *M. neoaurum* DSM 1381 originated from *M. neoaurum* ATCC 25790 by mutagenesis screening was reported to be able to accumulate HPD as the main product with a small portion of 4-HP [15]. Our previous research has led to the construction of an ideal 4-HP-producing strain (Δ*kstD1*), which is obtained by deleting *kstD1* in *M. neoaurum* DSM 1381. This Δ*kstD1* mutant could produce 14.18 g/L 4-HP from 20 g/L phytosterols in 168h [16]. However, the mechanism to accumulate 4-HP and HPD is rarely studied. In this study, the genome sequencings of *M. neoaurum* DSM 1381 and its parent strain *M. neoaurum* ATCC 25790, which have the ability to completely degrade sterols without accumulating any product, were carried out [15], and comparative genomic analysis was performed to identify the mutant site responsible for the accumulation of the C22 intermediate. With the comparative genomic analysis, the function and role of related genes were verified by the complement gene expression, and the effects of mutation sites on enzymes were discussed. These findings demonstrated that *M. neoaurum* DSM 1381 is an excellent target strain to discover the key enzymes of the steroid degradation pathway and provided some new insights into the accumulation of C22 steroids.

## 2. Results and Discussion

### 2.1. Sequencing and Gene Annotation of the Whole Genome of M. neoaurum DSM 1381 and M. neoaurum ATCC 25790

To analyze the presence of the steroid-related genes in *M. neoaurum* DSM 1381 and *M. neoaurum* ATCC 25790, the genomes were sequenced and annotated. As shown in Table 1 and Figure 1, their genome sizes are almost the same, with a total length of 5.58 Mb, which are similar to the genome sizes of *M. neoaurum* strains reported previously (*M. neoaurum* MN2 5.38 Mb, *M. neoaurum* NRRLB-3805 5.42 Mb, *M. neoaurum* ATCC 25795 5.47 Mb) [13,17]. There are 5232 and 5348 possible genes predicted in the genome of *M. neoaurum* DSM 1381 and *M. neoaurum* ATCC 25790, respectively. Moreover, the GC content in the gene region and intergenetic region is as high as 67% and 62.2%, respectively, which agrees well with previous reports [17,18]. Then the model strains *Mycobacterium* sp. VKM Ac 1815D and *M. neoaurum* ATCC 25795, whose whole-genome sequencings were completed previously, were selected for genome sequence alignment analysis and visualized [14] (Figure 1). The results show that, although they belong to the same genus, there is a huge number of differences among these genomes, among which *M. neoaurum* DSM 1381 is most similar to *M. neoaurum* ATCC 25790 while showing notable numbers of SNP, InDel, and deletions compared with the genomes of the *Mycobacterium* sp. VKM Ac 1815D and *M. neoaurum* ATCC 25795.

### 2.2. SNP Discovery Tree Analysis

Except for the two *M. neoaurum* strains mentioned above, four more strains were previously sequenced and deposited into the NCBI database, including *M. neoaurum* MN2, *M. neoaurum* MN4, *Mycobacterium* sp. VKM Ac 1816D, and *M. neoaurum* NRRL LB 3805. The characteristics of these strains were described briefly in the materials and methods section. *M. neoaurum* MN2 is an AD-producing strain, and *M. neoaurum* MN4 is mutated from *M. neoaurum* MN2 and has a higher AD yield [17]. To further confirm the genetic relationship between these strains, we conducted SNP discovery tree analysis. As shown in Figure 2, we found *M. neoaurum* ATCC 25790 and *Mycobacterium* sp. VKM Ac 1816D were clustered together as one branch that then clustered with *M. neoaurum* MN4, *Mycobacterium* sp. VKM Ac 1815D, and *M. neoaurum* NRRLB 3805. *M. neoaurum* DSM 1381 had the closest relationship with *M. neoaurum* MN2, followed by its origin strain *M. neoaurum* ATCC 25790, while the wild-type strain *M. neoaurum* ATCC 25795 had a farther relationship with other strains.

### 2.3. Prediction of Genes Related to Sterol Metabolism

In good agreement with previous reports [6,19,20], there is a gene cluster responsible for sterol metabolism in *M. neoaurum* DSM 1381 and *M. neoaurum* ATCC 25790. The well-studied strains, *M. neoaurum* ATCC 25795 and *Mycobacterium* sp. VKM Ac 1815D, were selected to annotate and analyze the homologs from the two newly sequenced strains, and the results were illustrated in Figure 3 [3]. The results were similar to those shown in the global comparison above. Gene deletions or insertions also occurred within the gene cluster within the same genus. Most of the identified genes, including KshA1, KshB1, KstD1, Hsd4A, KstR, KstR2, and mce4 operon, are present in the four strains’ gene clusters. Meanwhile, there are notable differences among the four clusters. For example, compared to *M. neoaurum* ATCC 25795, 10 genes were found inserted into the *mec4* operons of the other three strains. In the downstream region of Hsd4B from *Mycobacterium* sp. VKM Ac 1815D, a large fragment as long as 51.5 kb was unique relative to the other three genomes. In addition, there are three *KstDs* in the genomes of *M. neoaurum* ATCC 25795, *M. neoaurum* DSM 1381, and *M. neoaurum* ATCC 25790, but only one in *Mycobacterium* sp. VKM Ac 1815D. To sum up, these strains share a high identity as well as amounts of differences in sterol metabolic genes.

### 2.4. Mutation Analysis of Sterol Metabolism-Related Genes

To figure out the reasons for the phenotype differences between the two strains, the SNP and InDel were analyzed between the *M. neoaurum* DSM 1381 and its parent strain *M. neoaurum* ATCC 25790. Thirty-eight SNPs, including 28 nonsynonymous SNV and 10 synonymous SNV, and 25 InDels, in which 17 were located in the coding region and eight in the non-coding region, were found; the specific information of the mutants is shown in Appendix A. Two of the mutations appeared in the above-mentioned gene cluster and are located in the *kshA1*, which codes for 3-ketosteroid 9α-hydroxylase subunit A, and *hsd4A*, whose encoding protein has double-function of 17-hydroxytryptamine dehydrogenase and β-hydroxyacyl-CoA dehydrogenase, respectively [3,21,22,23]. The former is involved in the open-loop reaction of the B-ring and the latter is involved in the degradation of the C17 side chain.

As reported previously [24,25,26,27], the C17 side chain degradation process is similar to the β-oxidation cycle involved in lipid catabolism. The enzymes involved in this process are mainly oxidoreductases. All mutated genes, after excluding identified genes, are involved in other physiological processes and genes encoding hypothetical proteins; the remaining six genes, including *orf2435*, *orf2617-orf2616*, *orf187*, *orf1328*, *orf2235*, *orf1151*, were predicted to function as oxidoreductase and to be involved in lipid catabolism or sterol degradation. Moreover, *orf2188* was predicted to be a possible regulatory factor and had not yet been identified. The six mutated genes were selected to identify if they participate in sterol metabolism. To sum up, based on the gene annotation results, we selected eight mutated genes for further experiment to test whether they were involved in sterol side-chain degradation (Table 2).

### 2.5. The Functional Complement of the Selected Mutated Genes

The eight selected genes mentioned above, from *M. neoaurum* ATCC 25790, were cloned into pMV261. The resulting plasmids were transformed and expressed in *M. neoaurum* DSM 1381. The products from the sterol of the complemented strain were identified to test the complement results.

3-ketosteroid-9α-hydroxylase (KSH) is involved in the ring-opening oxidation of the steroid nucleus, which catalyzes the formation of 9α-hydroxyl-1,4-dienosteroid from 1,4-dien-3-one steroid [28], 9α-hydroxyl-1,4-dienosteroid are structurally unstable, leading to cleavage at the 9 and 10 positions of the B ring, disrupting the core structure of the steroid (Figure 3b). As shown in Figure 4, after the expression of the *kshA1* gene from *M. neoaurum* ATCC 25790 in *M. neoaurum* DSM 1381, the products 4HP and HPD were degraded without new 3-sterone compounds appearing, indicating that the N173D substitution destroyed the 3-ketosteroid-9α-hydroxylase activity of KshA1 which is responsible for steroid nucleus degradation. The expression of wild-type *hsd4A* leads to the transformation of HPD/4-HP to ADD/AD, which agrees well with the previously reported function [3], which is responsible for removing the last molecule propionyl CoA during the side chain degradation. In a word, the single base mutation in these two *kshA1* and *hsd4A* leads to the accumulation of 4-HP and HPD in *M. neoaurum* DSM 1381. In addition, there was no phenotypic difference after single-gene complementation of the other six mutated genes or co-expression with *hsd4A*, indicating they probably do not participate in sterol side-chain degradation.

### 2.6. The Structure and Sequence Analysis of Hsd4A

Hsd4A from ATCC 25795 was identified as a dual-function protein of 17-hydroxytryptamine dehydrogenase and β-hydroxyacyl-CoA dehydrogenase [3,21,22]. FabGs (β-oxoacyl reductases) are ubiquitous enzymes involved in fatty acid synthesis. The reaction entails NADPH/NADH-mediated conversion of β-oxoacyl-ACP (acyl-carrier-protein) into β-hydroxyacyl-ACP. The Hsd4A was rarely studied; however, in this study, we obtained the clue that the K171E mutation significantly damages the 17-hydroxytryptamine dehydrogenase activity of Hsd4A. To further figure out the possible reason, the homolog structure was built using SWISS-MODEL server [29] and AlphaFold2 [30,31].

Various Hsd4A homolog structures including a putative short-chain dehydrogenase from *Mycobacterium* smegmatis (4KZP) [32], a short-chain dehydrogenase from Mycobacterium avium (3QLJ), Human 17-beta-hydroxysteroid dehydrogenase Type 4 (1ZBQ), (3R)-Hydroxyacyl-CoA Dehydrogenase Domain of Candida tropicalis Peroxisomal Multifunctional Enzyme Type 2 (2ET6), (3r)-Hydroxyacyl-Coa Dehydrogenase Fragment Of Rat Peroxisomal Multifunctional Enzyme Type 2 (1GZ6) were resolved previously, and the identity is 79.5%/98%, 45.61%/94%, 41.67%/81%, 40.23%/80%, and 39.68%/81%, respectively. In this study, the amino acid sequences of these homologous fragments were compared with DSM 1381_Hsd4A (Appendix A). The 4KZP, 3QLJ, and 1ZBQ share a higher identity with target DSM 44074_Hsd4A. However, their functions were not identified. To obtain the homology model of DSM 1381_Hsd4A, the protein structure prediction algorithm AlphaFold2 was used to generate the model of the overall domain organization of DSM 1381_Hsd4A and the structural arrangements of the cofactor NAD^+^ bound in the cleft. Figure 5 shows the comparison of the simulated structures of the two Hsd4A proteins and the binding of the coenzyme NAD^+^ with them. The difference between the two proteins can be clearly observed; the distance between 171E residue of DSM 1381_ Hsd4A and NAD^+^ is 4.7 angstroms, which is obviously longer than the one from K171 residue of ATCC 25790_Hsd4A to NAD^+^, which is 3.8 angstroms. At the same time, Figure 5 shows the impact of K171E mutation on the overall configuration of Hsd4A protein.

Indicated by the structure analysis, K171 plays a crucial role in binding cofactor NAD^+^. Mutating the basic residue lysine to acidic residue glutamic acid destroys the inaction between the NAD^+^ with the NH3^+^. As a result, the function of Hsd4A was destroyed. In conclusion, the model we established has high reliability and can predict the structure of the Hsd4A well.

### 2.7. The Structure and Sequence Analysis of KshA1

KSH plays an important role during the degradation of steroids and is known to be a two-component Rieske oxygenase (Ro) system, consisting of a terminal oxygenase (KshA) and a ferredoxin reductase (KshB) [28,33]. The structure of KshA from *M. tuberculosis* and KshA1 as well as KshA5 from *Rhodococcus rhodochrous* DSM 43269 were solved previously; we used 2ZYL (3-ketosteroid-9α-hydroxylase from *M. tuberculosis*) as a template for homology modeling. Accordingly, the KshA monomer is arranged as a typical head-to-tail trimer [34]. KshA was identified as a flavoprotein reductase and two iron-sulfur proteins. KshA homologs have the typical Rieske Fe2S2 binding domain (C-X-H-X16,17-C-X2-H) and the nonheme Fe^2+^ motif (D-X3-D-X2-H-X4-H) [23,35,36]. As shown in Figure 6 and Appendix A, the Fe^2+^ motif region of KshA1_DSM 44074 appeared as D172-(NVT)-D-(MA)-H-(FFYV)-H184.

Combined with our experimental results, one of these SNPs was in a putative steroid-catabolism gene *kshA1*: an A517 nucleotide of *M. neoaurum* ATCC 25790 was substituted with a G nucleotide in *M. neoaurum* DSM 1381, thus resulting in the replacement of Asn173 with Asp. To explore the specific effect of the mutation site N173D on KshA, we used 2ZYL for homology modeling [34], and the amino acid sequence alignment is shown in Appendix A. According to previous reports, the mutation site N173D is located in the catalytic domain (amino acid residues 154–374) of KshA [34,36], which is consist with our predicted model. The mononuclear iron in *M. neoaurum* ATCC 25790is coordinated by His-179, His-184, Asp-302, and Asn-173, while the bidentate ligand Asn-173 in *M. neoaurum* DSM 1381 is replaced by Asp-173, which may disrupt the binding of the catalytic domain to the Fe, resulting in the inactivation of KshA1 and indicating the significance of N173 for the 3-ketosteroid 9α-hydroxylase function, which had not previously been mentioned and identified.

## 3. Materials and Methods

### 3.1. Chemicals, Strains, and Media

Prime STAR HS high-fidelity polymerase, Taq DNA polymerase, Kpn1, Sal1, EcoR1, and other restriction enzymes were purchased from TaKaRa Company; ClonExpress II One-Step Cloning Kit was purchased from Nanjing Vazyme Biotechnology Company; Genome Extraction Kit, Plasmid Miniprep kit, DNA fragment agarose gel recovery kit were purchased from Axygen; Kanamycin, Hygromycin were purchased from Shanghai McLean Biochemical Technology Co., Ltd. (Shanghai, China); ADD and AD standard were purchased from Yunnan Biological Products. Phytosterols (β-sitosterol 45%, campesterol 37%, stigmasterol 18%) was obtained from Jiangsu Yuehong Feed Co., Ltd. (Taizhou, Jiangsu, China); agar, yeast extracts, glycerol, and tryptone were purchased from Shanghai Macklin Biochemical Co., Ltd. (Shanghai, China) Glucose, Tween-80, ammonium salt, phosphate, methanol, ethyl acetate, n-hexane, etc. were purchased from Sinopharm Chemical Reagent Co., Ltd. (Shanghai, China)

*E. coli* was cultured with Luria–Bertani medium (LB medium) at 200 rpm, 37 °C. Mycobacterial competent cells were prepared and cultured in the medium including glycerol 20 g/L, (NH_4_)_2_PO_4_ 8 g/L, K_2_HPO_4_ 0.5 g/L, citric acid 3 g/L, ferric ammonium citrate 0.05 g/L, MgSO_4_ 7H_2_O 0.5 g/L, Tween-80 2 g/L. The medium used to culture mycobacterial cells was yeast extract 15 g/L, glucose 6 g/L, MgSO_4_ 7H_2_O 2 g/L, K_2_HPO_4_ 1.0 g/L, KNO_3_ 2.0 g/L, Tween-80 2 g/L, at pH 7.5–8.0. To perform steroid transformation the phytosterol 5 g/L and Tween-80 2 g/L were supplied, and 50 μg/mL kanamycin and 100 μg/mL hygromycin were added when needed.

### 3.2. Characteristics of the Used M. neoaurum Strains

*M. neoaurum* DSM 1381 is a mutant strain that was obtained by ultraviolet mutagenesis from wild-type strain *M. neoaurum* ATCC 25790 and is reported as a 4HP and HPD producer. *M. neoaurum* ATCC 25795 is a wild-type strain that is well studied. *Mycobacterium* sp. VKM Ac-1815D is the well-known producer for AD with a molar yield of 68–72% [37], but is also able to accumulate ADD (6–10%), 20-hydroxymethyl pregn-4-ene-3-one (HMP) (14–16%), and a small amount of 20-hydroxymethyl pregna-1,4-diene-3-one (HMPD). The main product of *Mycobacterium* sp. VKM Ac-1816D is ADD (70–72%), and AD (2–4%), HMPD (14–16%), and a small amount of HMP are by-products [14]. *M. neoaurum* NRRLB 3805 is another AD producer with less by-product ADD compared with *Mycobacterium* sp. VKM Ac-1815D. *Mycobacterium neoaurum* MN2 is a well-characterized AD producer which was obtained from soil organisms. *M. neoaurum* MN4 is a mutated strain from *M. neoaurum* MN2 with a higher AD yield and an increased tolerance to phytosterols (Table 3).

### 3.3. Genome Sequencing and Annotation

The genomes of *M. neoaurum* DSM 1381 and *M. neoaurum* ATCC 25790 were isolated using the DNA extraction kit (Axygen) according to modified manufacturers’ protocols, which are described briefly as follows: after the collection with centrifuges, the *M. neoaurum* pellets were resuspended with 150 µL Buffer S (RNase A added) and treated with 20 µL 5 mg/mL lysozyme at 37 °C for 30 min. After adding 30 µL 0.25 M EDTA, the reaction was ice-incubated for 5 min. Then 450 µL Buffer GA was mixed and incubated at 65 °C for 30 min. The subsequent procedures were processed strictly according to the manufacturers’ protocols. After certifying the DNA qualification with 1% agarose gel electrophoresis and nanodrop (Thermo Scientific), the genome was sequenced by Majorbio (Shanghai). The whole-genome sequencing was performed with Illumina Hiseq. The raw data was first quality-trimmed, and then assembled with SOAPdenovo v2.04 (http://soap.genomics.org.cn/, accessed on 20 November 2020) and adjusted with GapCloser v1.12. Then the assembled genome sequences were deposited in the National Center for Biotechnology Information (NCBI) database under accession number CP115600 and CP115466, respectively.

Genes of the assembled genomes were predicted with the software Glimmer 3.02 (http://www.cbcb.umd.edu/software/glimmer/, accessed on 25 November 2020) [39,40]. The predicted protein sequences were blasted using BLAST 2.2.28+ against the databases including Nr, genes, string, and GO.

### 3.4. Whole Gene Alignment Analysis and Visual Display

The annotation results were visualized by CGV software. There are four *M. neoaurum* strain genomes selected for sequence alignment: the genome sequencing results of *M. neoaurum* DSM 1381, *Mycobacterium* sp. VKM Ac 1815D, *M. neoaurum* ATCC 25790 and *M. neoaurum* ATCC 25795. The graphical view of the alignments was rendered using BLAST Ring Image Generator (BRIG).

### 3.5. SNP and Indel Analysis

To detect the single-nucleotide polymorphisms (SNPs), the reliable raw short read sequences of *M. neoaurum* ATCC 25790 obtained were compared to assembled genome sequences of *M. neoaurum* DSM 1381 with the BWA package. The results were analyzed with Samtools for significance values of each genome nucleotide. The calling results were checked out manually by aligning the two assembled genomes with the NCBI BLASTN SUITE. In the end, Annovar was employed to annotate the found SNPs and Indels [41].

### 3.6. Consensus Parsimonious Tree

kSNP 3.0 was used to scan SNPs across 18 genomes and to build a consensus parsimonious tree. One hundred equivalent parsimonious trees were constructed, and support values and consensus trees were derived from them.

### 3.7. Comparative Analysis of Steroid Degradation-Related Gene Clusters

Blast alignment was performed on the two groups of gene sequences, and the alignment results with high similarity (Similarity > 90%) were selected. The relationship between the alignment results, gene names, and annotation information was sorted out, and the graph was drawn by script.

### 3.8. Functional Complementation of Differential Genes

The complete coding sequences of hsd4A, kshA1, orf624, orf1123, orf1296, orf2188, and orf2389, amplified from the *M. neoaurum* ATCC 25790 genome, were inserted in the EcoRI site of pMV261. To perform functional complementation, the resulting plasmids were transformed into the mutant strain *M. neoaurum* DSM 1381 according to the procedure described previously [42]. The plasmid pMV261hyg-hsd4A was constructed based on pMV261-hsd4A to get co-expression of hsd4A with *orf624*, *orf1123*, *orf1296*, *orf2188*, and *orf2389*. The Hyg expression cassette was obtained from pSBY3_ligD-HygR and inserted in the SspI site on the Kan coding sequences. Then, pMV261hyg-hsd4A was introduced into DSM1381_orf624, DSM1381_orf1123, DSM1381_orf1296, DSM1381_orf2188, and DSM1381_orf2389 and selected with Hyg and Kan.

### 3.9. Analysis of Sterol Fermentation Products

#### 3.9.1. Sample Processing

When conducting sterol fermentation experiments with *M. neoaurum*, samples were taken every 24 h. The fermentation samples were repeatedly extracted three times with an equal amount of ethyl acetate. The three extracts were mixed, filtered through a 0.22 μm membrane, and 75 μL of the filtrate was mixed with an equal volume of squalene (Sigma, Waltham, MA, USA) internal reference compound solution for gas chromatography detection. The procedure was to take 50 μL of the filtrate to evaporate the solvent, dissolve it with an appropriate amount of methanol ultrasonically, and carry out liquid chromatography detection.

#### 3.9.2. TLC Detection Method for Steroids

TLC can be used as a qualitative detection method with ethyl acetate/n-hexane (6:4) as the developing solvent. Among them, 3-sterone compounds can be observed under ultraviolet irradiation, while the color development of sterols needs to be observed by spraying with 20% dilute sulfuric acid and then treating at 115 °C for 20 min.

#### 3.9.3. The Detection Method of 3-Sterone Compounds

The 3-sterone compound has an absorption peak at 254 nm, so it was quantitatively detected at 254 nm by high-performance liquid chromatography. Chromatographic analysis conditions: C18 reversed-phase chromatography column (Agilent Extend-C18 column, 4.6 × 250 mm, 5 μm (Agilent, Santa Clara, CA, USA)); mobile phase: methanol/water (80:20, *v*/*v*); flow rate 0.8 mL/min; column temperature 40 °C; injection volume was 20 μL. The procedure was to take 50 μL of the treated product extract in a ventilated place to evaporate cleanly, and then to dissolve it with an appropriate amount of anhydrous methanol ultrasonically.

#### 3.9.4. The Quantitative Detection Method of Sterols

Gas chromatography methods were used for the quantitative analysis of phytosterols. Gas chromatography analysis method: gas chromatography column (Agilent Rtx-5, 30 m × 0.53 mm × 5.0 µm), inlet temperature 320 °C, column temperature 300 °C; detector temperature 320 °C. Squalene (Sigma) was used as an internal reference compound.

### 3.10. Prediction of the Three-Dimensional Structures of the Hsd4A and KshA1 Enzymes

The structural models of the Hsd4A and KshA1 enzymes were predicted with AlphaFold2 using online resources and the protocol described by Jumper et al. [31]. To conduct the AlphaFold2 prediction, the Hsd4A and KshA1 enzyme sequences were input into the jupyter notebook for ColabFold [43]; then, the AlphaFold2 models of Hsd4A and KshA1 were obtained using as input the amino acid sequence and a HMMer [30] multiple sequence alignment (default method from Deepmind) [31]. The AlphaFold2 built an end-to-end network to optimize the final model [44]; furthermore, AlphaFold2 used attention modules to derive distance constraints and built structural models from them with 3D equivariant transformer neural networks [45,46], which operate directly on atoms in three-dimensional space. The prediction also contained side-chain information.

## 4. Conclusions

C22 steroid intermediates, including 4-HP, HPD, 1,4-HP, 9-OH-4-HP, are products of an incomplete degradation of steroid side chains. Compared with C19 steroid intermediates, C22 intermediates are more suitable for the synthesis of progesterone and adrenocortical hormones [3]. For example, the C22 intermediate 9-hydroxy-3-oxo-Pregna-4,17 (20)-diene-20-carboxylic acid was reported to be a precursor to generate 9-hydroxy-4, 16-pergnadiene-3, 20-dione under the catalysis of H_2_O_2_ in the presence of MoO_4_^2-^, and the latter can be used to synthesize corticosteroid hormones [47]. Development of C22 steroid-producing strains continues to interest researchers. For instance, Xu found that the formation of C22 intermediate 4-HP is related to Hsd4A [3], which started the molecular research of the C22 intermediate production strain [27,48,49]. However, due to the complex pathway of steroid-degradation and the unclear accumulation mechanism of C22 intermediates, only a few industrial C22 steroid-producing strains have been developed, and the current strains retain some drawbacks, including low product purity and low molar yield. The key genes and enzymes involved in the steroid-degrading pathway in *M. neoaurum* have drawn more attention and have been selected as the origin strain to construct the admirable producers for important steroid intermediates. To research on the steroid degradation mechanism of *M. neoaurum*, we carried out the genome sequencing, gene annotation, and comparative genomic analysis of HPD/4-HP accumulation mutant *M. neoaurum* DSM 1381 and its parent strain *M. neoaurum* ATCC 25790. The sterol degradation-related genes were predicted and studied, and the key genes *hsd4A* and *kshA1* responsible for the phenotypic difference of *M. neoaurum* DSM 1381 were identified by functional complementation.

According to the genome sequencing results, the GC content of the two genomes is more than 65% and the genomes’ size is around 5 Mb, which is consistent with the reported data. As shown in Figure 1, the two genomes share a high identity with the genome of wild type *M. neoaurum* ATCC 25795 and the AD-producer *Mycobacterium* sp. VKM Ac 1815D in most genomic regions. However, there are also plenty of low non-identical regions, deletions, and inserts found both inside and outside the sterol metabolism-related gene clusters. For example, the long insert observed in the gene cluster for steroid degradation is as long as 51.5 kb. Consistently, the *M. neoaurum* DSM 1381 shares a far genetic distance with *M. neoaurum* ATCC 25795, as shown in the SNP evolutionary tree (Figure 2). The numbers of genetic differences make it hard to locate the mutant sites responsible for the phenotype differences between these strains. Thus, SNP/InDel analysis was only performed between *M. neoaurum* DSM 1381 and its parent strain, which is more feasible to discover the key genes. Considering gene annotation and SNP/InDel results, eight mutated genes were selected as possible sites that lead to the accumulation of 4-HP and HPD in *M. neoaurum* DSM 1381.

Then the gene complementation experiment was processed to confirm the key mutation sites. The complementation of *hsd4A* led to the generation of AD and ADD, and the productions of 4-HP and HPD were almost undetectable, proving that *hsd4A* was involved in the removal of the last molecule of propionyl-CoA in the process of side-chain degradation and that 171Lys of Hsd4A is a key residue for enzyme activity. Loss of protease activity is the reason that *M. neoaurum* DSM 1381 accumulated HPD/4-HP rather than ADD/AD. In order to gain a deeper understanding of the effect of this mutation, we also performed homology modeling on Hsd4A. We proposed that the replacement of basic amino acids with acidic amino acids affects the previous inaction between NH3^+^ and NAD, thus destroying the enzymatic activity. This study explored whether the inactivation of DSM 1381_hsd4a led to the inhibition of the accumulation of C19 intermediates during the transformation of phytosterol in *M. neoaurum* DSM 1381, leading to the incomplete degradation pathway, which explained why the main accumulation product HPD and the secondary accumulation product 4HP of *M. neoaurum* DSM 1381 were both C22 intermediates.

Theoretically, when the target genes are supplied, the 4HP and HPD will be totally degraded. Thus, there must be another gene responsible for AD/ADD degradation that was inactive. Indeed, after expressing *kshA1* from *M. neoaurum* ATCC 25790 in *M. neoaurum* DSM 1381, the 4-HP and HPD were no longer observed, and no new 3-sterone compounds were generated, indicating that the 3-ketosteroid-9α-hydroxylase, which is responsible for the ring-opening reaction, is inactivated in *M. neoaurum* DSM 1381. The amino acid substitution caused by the point mutation A517G in KshA1 does lead to the loss of KshA1 activity, which is a prerequisite for the accumulation of steroid-containing intermediates and the production of HPD/4HP. Similarly, we performed homology modeling on KshA1 and found that the N173D may lead to changes in the coordination of key amino acids in the catalytic domain with the mononuclear iron. This study explored whether the inactivation of DSM 1381_KshA1 causes no production of 9 α- hydroxylation products, which retain the integrity of the steroid nucleus and provide a prerequisite for the accumulation of HPD and 4HP in *M. neoaurum* DSM 1381.

In summary, we sequenced the genomes of two *M. neoaurum* strains. The resultant information and analysis lead to the discovery of two crucial residues related to C22 steroid accumulation. We demonstrated this is a promising way to discover the key enzymes on the steroid degradation pathway. Especially, there are plenty of steroid intermediate-producing strains developed using mutation and screen methods previously, and the genome sequencing cost is decreasing dramatically. The genome information obtained in this study is also useful as a reference for the strain development to reduce the side products and increase productivity and product purity.

## Figures and Tables

**Figure 1 ijms-24-06148-f001:**
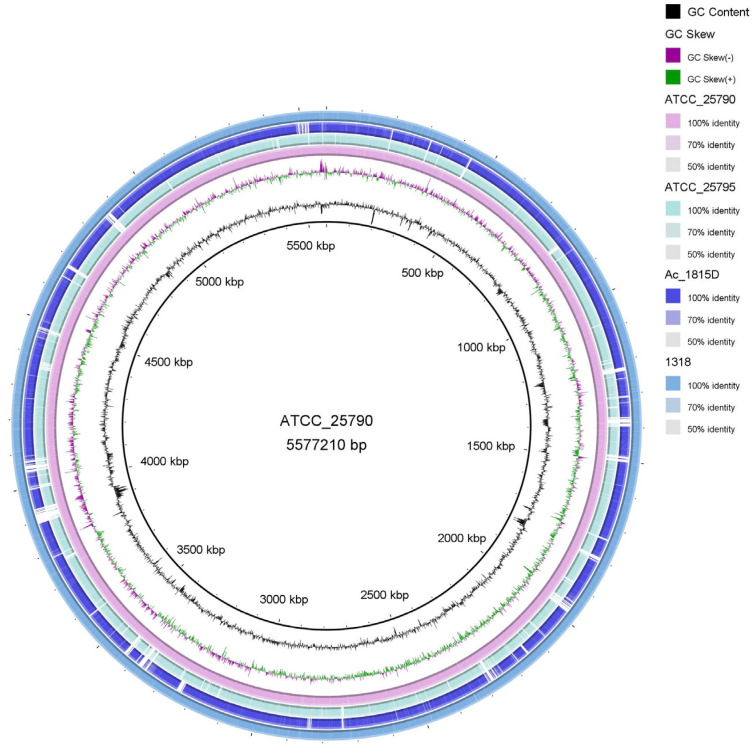
Global comparison of *M. neoaurum* DSM 1381. *M. neoaurum* ATCC 25790genomes, *M. neoaurum* ATCC 25795, and *Mycobacterium* sp. VKM Ac 1815D sequence.

**Figure 2 ijms-24-06148-f002:**
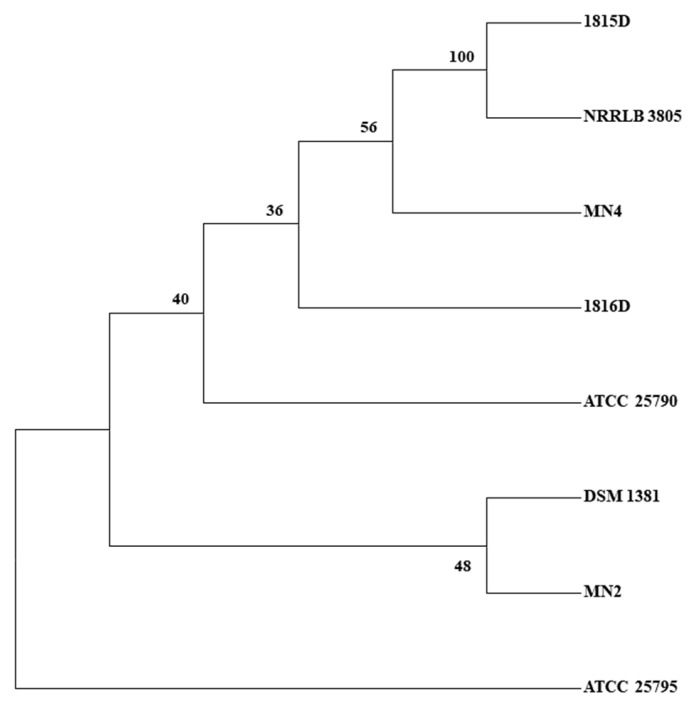
Consensus parsimonious tree derived from SNPs detected in eight strains of *M. neoaurum.* DSM 1381, *M. neoaurum* DSM 1381; ATCC 25790, *M. neoaurum* ATCC 25790; ATCC 25795, *M. neoaurum* ATCC 25795; VKM Ac 1815D, *Mycobacterium* sp. VKM Ac 1815D; VKM Ac 1816D, *Mycobacterium* sp. VKM Ac 1816D; MN2, *M. neoaurum* MN2; MN4, *M. neoaurum* MN4; NRRLB 3805, *M. neoaurum* NRRLB 3805.

**Figure 3 ijms-24-06148-f003:**
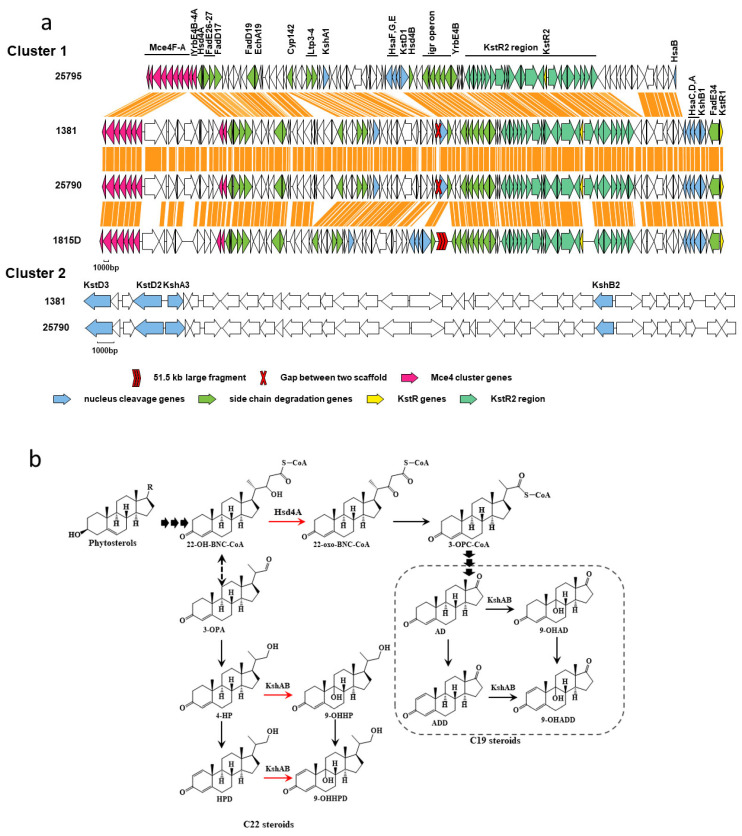
Partial gene cluster encoding the catabolism of sterols in *M. neoaurum* DSM 1381and putative phytosterol metabolic pathway by *M. neoaurum* DSM 1381. (**a**) Genes in the map are color-coded according to assigned or proposed function: Orange, *mce* cluster genes for steroids transportation; Green, side-chain degradation genes; Blue, nucleus cleavage genes; Light green, the genes under the control of KstR2; Yellow, the gene encoding the transcriptional repressor KstRs; (**b**) The red arrow indicates the step blocked in *M. neoaurum* DSM 1381.

**Figure 4 ijms-24-06148-f004:**
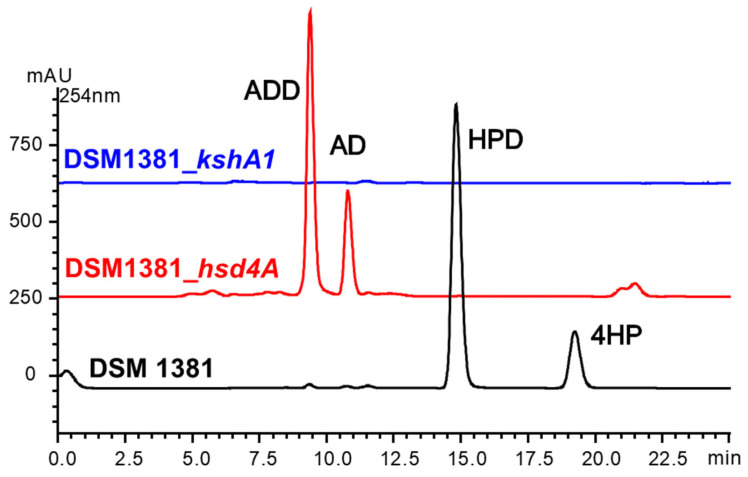
HPLC chromatogram comparison of the products from the transformation of phytosterols by strains DSM 1381_hsd4A (red), DSM 1381_kshA1 (blue), and *M. neoaurum* DSM 1381 (black).

**Figure 5 ijms-24-06148-f005:**
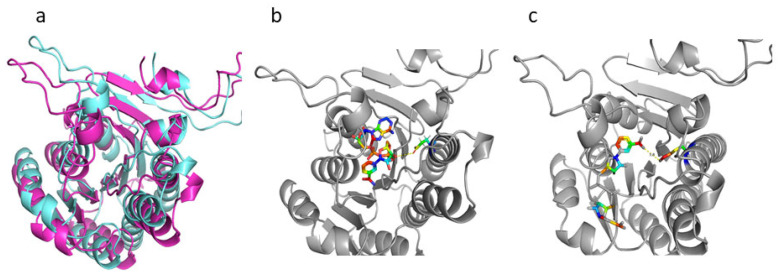
Structural model of Hsd4A. (**a**) DSM 1381_Hsd4A, aquamarine; ATCC 25790_Hsd4A, light magenta; (**b**) DSM 1381_Hsd4A, gray, 171E and NAD+ shown in sticks; (**c**) DSM 1381_Hsd4A, gray, K171 and NAD+ shown in sticks.

**Figure 6 ijms-24-06148-f006:**
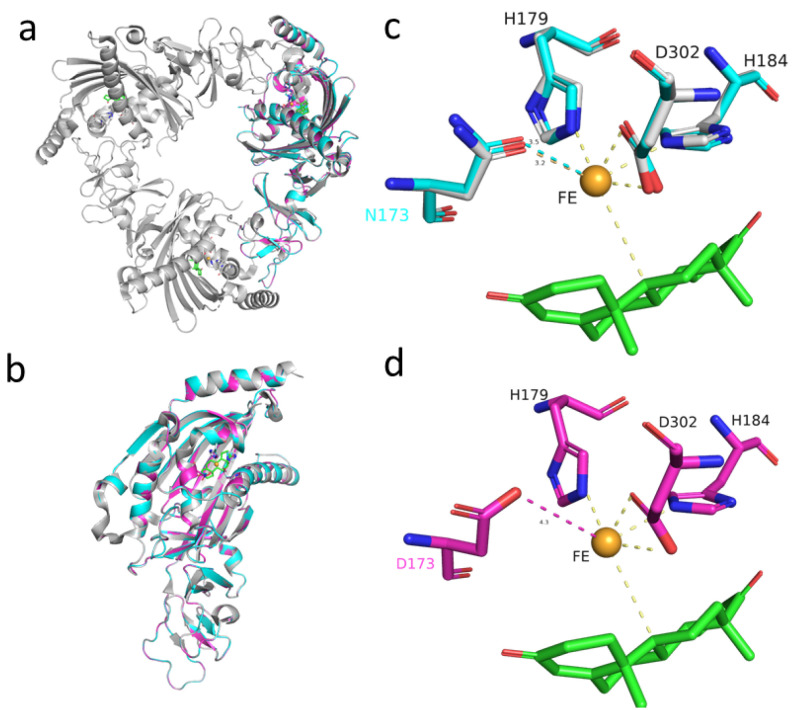
Structure model of KshA1. DSM 1381_ KshA1, pink; DSM 44074_KshA1, blue; structural template, gray; substrate AD, green; the non-heme Fe2+, gold (**a**,**b**). The residues within 4 Å of Fe are shown in sticks (**c**,**d**).

**Table 1 ijms-24-06148-t001:** Results of genome sequencing and gene annotation.

Strains	DSM 1381	ATCC 25790
Gene num	5232	5348
Gene total length	5,577,916 bp	5,577,210 bp
Gene average length	971 bp	951 bp
Gene density	0.937 genes per kb	0.958 genes per kb
GC content in gene region (%)	67.2	67.2
Gene/Genome (%)	91.1	91.3
Intergenetic region length	494,836 bp	487,875 bp
GC content in the intergenetic region (%)	62.2	62.2
Intergenetic length/Genome (%)	8.97	8.74

**Table 2 ijms-24-06148-t002:** The mutated genes in *M. neoaurum* DSM 1381 predicted to be responsible for steroid degradation.

Description of the Gene Product	Gene ID		SNV		aa Mutation	
Nonsynonymous SNV	ATCC 25790	DSM 1381	ATCC 25790	DSM 1381	ATCC 25790	DSM 1381
KshA1	807	14	517A	517G	173Asn	173Asp
Hsd4A	775	46	511A	511G	171Lys	171Glu
NAD(P)/FAD-dependent oxidoreductase	1328	1296	1025C	1025G	342Ala	342Gly
FadR family transcriptional regulator	2235	2188	446T	446A	149Leu	149Gln
SDR family NAD(P)-dependent oxidoreductase	1151	1123	upstream 34G	upstream 34T		
Indel	ATCC 25790	DSM 1381	Indel in DSM 1381		Mutation start position	
2,4-dienoyl-CoA reductase	2435	2389	insert, 4 bp		660	
acyl-CoA synthetase	2617–2616	2573	gap, 34 bp		807	
acyltransferase	187	624	gap, 1 bp		461	

**Table 3 ijms-24-06148-t003:** Information on the strains used for comparative analysis.

Strain	Abbreviation	Main Product	Side Products	Reference
*M. neoaurum* DSM 1381	DSM 1381	HPD	4HP	[15]
*M. neoaurum* ATCC 25790 (*M. neoaurum* ATCC 25790)	ATCC 25790	NA	NA	[15]
*M. neoaurum* ATCC 25795	ATCC 25795	NA	NA	
*Mycobacterium* sp. VKM Ac-1815D	Ac 1815D	AD	ADD, HP, HPD	[37]
*Mycobacterium* sp. VKM Ac-1816D	Ac 1816D	ADD	AD, HP, HPD	[14]
*M. neoaurum* MN2	MN2	AD	ADD	[17]
*M. neoaurum* MN4	MN4	AD	ADD	[17]
*M. neoaurum* NRRLB 3805	NRRLB 3805	NA	NA	[38]

## Data Availability

The data supporting these findings can be found in the Appendix A.

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
