# Peer review of "Whole-Genome Analysis of Mycobacterium neoaurum DSM 1381 and the Validation of Two Key Enzymes Affecting C22 Steroid Intermediates in Sterol Metabolism"

_ijms, 2023, doi:10.3390/ijms24076148_

Round 1

Reviewer 1 Report

Mycobacterium neoaurum DSM 1381 degrades sterols and accumulates C22 intermediates that is valuable precursors for steroid drugs. The strain DSM 1381 was originally generated from the parent strain DSM 43536 that completely degrade sterols without accumulating the intermediates. In this manuscript, the authors determined whole genome sequences of those strains, and performed comparative genomic analysis, identifying nonsynonymous SNVs, coding and non-coding Indels. Furthermore, the authors further expressed the wild type genes, kshA1 and hsd4A, from the parent strain into the mutant strain, and showed that 173N of KshA1 and 171K of hsd4A are indispensable for the function of those proteins. Finally, computational models were built to explain the role of mutations.  Currently numerous amounts of genome sequence data have been accumulated from various organisms including bacteria. However, the actually connection between genome sequence data and actual phenotype is not demonstrated in most of case. The authors’ work is a good example of application of comparative genome sequence data to find the biological function of genes. Also, their work is important to understand degradation mechanisms of sterols, and the experiments were nicely done. However, I have been the following suggestions that should be addressed.

1. Substantial editing of English language is required throughout the manuscript.

2. Lines 105-107: This part needs to be removed from the manuscript

3. The name of bacteria should be italicized throughout the text.

4. It might be useful for readers if the authors could provide a figure to explain the degradation pathway by Mycobacterium.

5. Two different names, DSM 43536 and ATCC 25790 are used to describe same strain, and it is confusing. The authors should use one of them throughout the manuscript.

Reviewer 2 Report

The manuscript intitled “Whole-genome analysis of Mycobacterium neoaurum DSM 1381 and the validation of two key enzymes affecting C22 steroid intermediates in sterol metabolism” deals with the comparison of M. neoaurum DSM 1381 and its parent strain M. neoaurum DSM 43536 genomes to understand the mechanism of C22 steroid accumulation. The authors have overexpressed KshA1 and Hsd4A in M. neoaurum DSM 1381 and found out that 173N of KshA1 and 171K of Hsd4A are indispensable to maintain their activity. They have performed amino acid sequence alignment analysis and 3D models of these two enzymes. With these analysis they could explain the reasons for the accumulation of C22 intermediates in M. neoaurum DSM 1381. This study is of great importance because C22 intermediates are highly valuable precursors in steroid drugs production. Biotransformation methods are extremely advantageous and several studies like this one are necessary to increase its use instead of Chemical synthesis.

The manuscript is clear, well written and conclusions are supported by results. Methodology is adequate.

Abstract: Some abbreviations should be identified, for example SNVs, N173D, and K171E. The abstract text is too long. Avoid introductions to the subject in this part.

Introduction: should be separated in more paragraphs.

Specific comments:

Lines 75-78: missing reference

Lines 105-107: text must be excluded.

Line 114: italics for microbial species name

Figure legend: must follow template format.

Subtitles in italics: microbial species name should be underlined.

Line 371: correct mistake

Abbreviation for minutes is “min”, not “mins”. Correct all over the text.

Standardize space between paragraphs and the space between paragraphs and subheadings. Follow template format.

Be aware to use space between words and [].

“Disclaimer/Publisher’s Note:” should be in line 542 and remove the space (pages in blanc)

Revise these sentences:

“Mycobacterium neoaurum DSM 1381 ... is a strain of degrading phytosterols and accumulating important C22 steroid intermediates,...”

“Steroid drugs, have a variety of pharmacological and physiological activities, are widely used in anti-inflammatory, anti-tumor, regulating sexual ability and birth control...”

“Agree well with previous reports [6,20,21], there is a gene cluster...”

“Take another 50 μL of the filtrate to evaporate the solvent...”
